# Gradient Printing Alginate Herero Gel Microspheres for Three-Dimensional Cell Culture

**DOI:** 10.3390/ma15062305

**Published:** 2022-03-20

**Authors:** Youping Gong, Honghao Chen, Wenxin Li, Chuanping Zhou, Rougang Zhou, Haiming Zhao, Huifeng Shao

**Affiliations:** 1School of Mechanical Engineering, Hangzhou Dianzi University, Hangzhou 310018, China; gyp@hdu.edu.cn (Y.G.); chenhonghao@hdu.edu.cn (H.C.); wwenxindiaolong@163.com (W.L.); 2State Key Laboratory of Fluid Power and Mechatronic Systems, School of Mechanical Engineering, Zhejiang University, Hangzhou 310027, China; zhaohaiming198108@163.com; 3Jiangsu Key Laboratory of 3D Printing Equipment and Manufacturing, Nanjing Normal University, Nanjing 210042, China

**Keywords:** sodium alginate, heterogeneous gel spheres, biological manufacturing, three-dimensional cell culture

## Abstract

Hydrogel microspheres are widely used in tissue engineering, such as 3D cell culture and injection therapy, and among which, heterogeneous microspheres are drawing much attention as a promising tool to carry multiple cell types in separated phases. However, it is still a big challenge to fabricate heterogeneous gel microspheres with excellent resolution and different material components in limited sizes. Here, we developed a multi-channel dynamic micromixer, which can use active mechanical mixing to achieve rapid mixing with multi-component materials and extrude the homogenized material. By changing the flow rate ratio of the solutions of the two components and by rapidly mixing in the micromixer, real-time concentration change of the mixed material at the outlet could be monitored in a process so-called “gradient printing”. By studying the mixing efficiency of the micromixer, its size and process parameters were optimized. Using the novel dynamic gradient printing method, the composition of the hydrogel microspheres can be distributed in any proportion and alginate heterogeneous gel microspheres with adjustable cell concentration were fabricated. The effects of cell concentration on cell viability and proliferation ability under three-dimensional culture conditions were also studied. The results showed that cells have very low death rate and can exchange substances within the microspheres. Due to the micromixing ability of the micromixers, the demand for biological reagents and materials such as cells, proteins, cytokines and other materials could be greatly reduced, which helps reduce the experimental cost and improve the feasibility of the method in practical use. The heterogeneous gel microsphere can be greatly valuable for research in various fields such as analytical chemistry, microarray, drug screening, and tissue culture.

## 1. Introduction

In organisms, there are universal heterogeneity from DNA molecules to cells, individuals and finally tissues and organs. Especially in biological 3D printing technology, the heterogeneity of in vitro artificial tissue is particularly important. It is not only the basic characteristic of cell microenvironment construction, but also the difficulty of customized biological printing technology. The heterogeneity of Gel Microspheres mainly includes spatial patch and gradient. In addition, the distribution of components in Gel Microspheres and the spatial distribution in spheres were emphasized, while the gradient was mainly expressed as the concentration difference of components in spheres. In human tissues, the types of cells, the arrangement direction of cells, the distribution and concentration of cytokines and the distribution and difference of matrix materials are not homogeneous, and the distribution law is very complex and changeable. The reasonable construction of heterostructures will be used in the research of tissue engineering, material properties, cell activity mechanism and drug release in vitro [1,2,3,4,5,6,7,8,9,10]. At present, with the crossing of disciplines and more and more researchers engaged in bio manufacturing, a large number of new technologies have emerged in the field of heterogeneous microspheres, including (i) the method based on microfluidic control; (ii) the method based on coaxial fluid shear; and (iii) the method based on the presence of template. These heterogeneous microspheres can be used as heterogeneous cell carriers, drug release units, biological probes and actuators for biomedical applications in the biological field [11,12,13,14,15].

For the preparation of microspheres based on microfluidic control, including hot phase [16], PDMS and UV photocurrent combination [17], cross microfluidic chips simultaneously prepare multiple gel beads, and prepare the gel ball [18,19,20,21] by magnetic field method. The core is to control the dispersion of equivalent fluid flow to achieve heterogeneous gel beads. Microfluidic method can only prepare microspheres with relatively small viscosity, and cannot accurately control the composition distribution in microspheres. The coaxial fluid shear method uses coaxial conical capillaries to build equipment. As a dispersed phase, the gel material is cut into droplets under the circumferential coaxial fluid shear flow. The prepared microspheres have core-shell structure and are especially suitable for microencapsulation. At the same time, through the connection of multi-level coaxial devices, various core-shell packages can be prepared [22,23,24,25,26]. Microencapsulation of cells using coaxial microfluidic nozzle can stain encapsulated live/dead cells, and can also be used to simulate the expression and release of proteins. By connecting multistage coaxial microfluidic devices, more complex multi-core heterogeneous microspheres can be prepared [27,28]. Template method is a very simple way to prepare gel microcarriers. In this method, a mold for casting is first prepared. In biological research, it is very common to prepare PDMS mold by soft etching. After that, the solidified hydrogel is deposited on the mold and then solidified by ultraviolet irradiation or by other means. By this simple method, high uniformity gel microcarriers can be produced in batches in a short time. Template method can be used to prepare gel particle array for transfer reagent to absorb or release small molecule chemical reagents. Compared with traditional micropipette method, the method has higher efficiency and controllability [29,30]. Through the comparative study of printing technology of these heterogeneous gel microspheres, we can find that over the past few years’ development and exploration, heterogeneous gel microcarriers have attracted more and more attention, especially in the field of biological printing and drug release. The structure of heterogeneous gel microspheres has evolved from the original single core shell structure or partition structure to more and more complex shapes, and the manufacturing process has become more and more abundant. However, it is not a simple matter to construct complex structures and multiple components within tiny gel spheres. Due to the particularity of cell printing, there are not many kinds of hydrogels that can be used as carriers. The formation of heterostructures in microspheres is mainly caused by laminar flow, phase separation and light curing.

In order to achieve this heterogeneous spatial structure, the gel materials used need to be satisfied: (1) the interface can be distinguished from heterogeneous components with certain adjustable fluid viscosity. (2) it has good biocompatibility, biodegradability and permeability. In order to meet the functions of cell survival and proliferation in the gel ball, the bio gel needs the similar biological properties of extracellular matrix. (3) gel has certain mechanical strength to meet the requirement of microspheres in vitro and in vivo, and can achieve an adjustable biodegradation cycle.

This paper presents an improved dynamic hybrid injection gradient gel method, which adopts active mixing principle, moving parts as a biocompatible blender, and rapidly mixing different components of alginate gel in the mixing chamber, and then through the replacement needle installed at the outlet of the mixing chamber to achieve homogeneous material extrusion. By changing the ratio of two-component solution to liquid flow rate and rapid mixing through micro mixer, the real-time concentration change of mixed material at the outlet, i.e., gradient printing is realized. The paper includes the following aspects: (1) Preparation of heterogeneous gel spheres containing material and gridient printing detailed method; (2) The simulation model of Heterogeneous materials stirring process simulation with different parameters was constructed to help choose the optimal parameters of the printing process. (3) The Heterogeneous Gel Microspheres were printed on the platform which has dynamic stirring extrusion sparger with different parameters, and these experimental results will further verify can efficiently fabricate various gradient structured Gel Microspheres.

## 2. Materials and Methods

### 2.1. Materials

In order to obtain heterogeneous gel spheres containing cells, microspheres as heterogeneous components can have proteins, cells and other materials, and alginate solution is the basic component. In order to make the alginate condensate ball react with calcium chloride solution sufficiently without causing the cells to be removed from the culture environment for too long to die, the gel beads will be placed in 2% calcium chloride solution for a short time to fully cross link them. In order to verify that gel microspheres can be used for cell culture, chondrocytes were cultured in vitro and verified in subsequent experiments. Biological grade sodium alginate powder is non-toxic and odorless white powder with very high hydrophilicity. It can be mixed with an aqueous solution with pH values greater than 3, from which a homogeneous solution with high dynamic viscosity is formed [31,32,33,34]. A summary of each name, specification and company of the reagent is provided in Table 1.

### 2.2. Preparation of Alginate Gel Microspheres

The preparation process of alginate heterogeneous gel microspheres can be divided into three steps: material mixing, droplet ejection and droplet crosslinking. In order to obtain heterogeneous gel balls containing cells, the core component is printing nozzles. The specially designed nozzle can realize the controllable size of microspheres, the uneven distribution of different components and the three-dimensional wrapping of cells. Two injection ports symmetrically distributed on both sides of the mixing chamber are, respectively, filled with pure sodium alginate solution and sodium alginate cell mixture. The flow ratio of pure alginate solution and the mixing chamber nozzle of cell suspension are controlled by micro jet pump and the flow is accurately controlled. In the process of injecting solution, the liquid volume ratio of two different components is changed in real time, and the stepping motor is turned on to drive the mixing rotor to mix the two solutions evenly, so as to realize the real-time change of the concentration of mixed materials at the outlet. A micron dispensing needle is installed at the outlet of the mixing chamber, as shown in Figure 1. The six hole plate is divided into two areas, printing area and transition area. The printing area is used to receive heterogeneous gel microspheres containing cells. The transition zone is used to receive excess waste, pre extruding materials and gel beads of transitional concentration. A cell culture medium containing 2% calcium chloride was added to the printing area. Cells containing sodium alginate gel solution were obtained in ideal mixing state. Open the compressed air button. Under the driving of the gas pressure, a spherical droplet is formed at the dispensing needle. When the gel drops into printing area, then it is crosslinked with calcium chloride to form heterogeneous gel beads with different cell concentration in different parts.

### 2.3. Gel Microsphere Manufacturing Device

To obtain heterogeneous gel ball-encapsulated cells, the core component lies in the printing nozzle. This paper proposes an improved biological dynamic mixer nozzle, which adopts active mixing principle as follows in Figure 2III. The biocompatible agitators were used as the moving parts, and the extrusion of homogeneous material was achieved using a replaceable needle installed at the outlet of the mixing chamber. Additionally, a push rod injection pump was applied as the feeding system, and the linear motion of the injection pump was transformed into the liquid flow rate of the gel solution through a single chip microcomputer; and the flow feed was precisely controlled. In addition, the real-time concentration change of the mixed material at the outlet, i.e., gradient printing, was achieved by changing the ratio of the two components to the liquid flow rate and by rapid mixing through the micro-mixer. The micro-mixer was mounted directly onto the three-dimensional motion platform as a printer nozzle. The replaceable metal Ruhr joint and the main body of the mixing chamber were connected by threads to monitor the plug-in and pull-out of the external liquid intake pipe. Figure 2I is the physical structure of the gel microsphere preparation device. Figure 2II is the internal detailed structure of the gel microsphere preparation device. The main body of the mixing chamber was divided into two parts: the stirrer and the stepper motor shaft, which were separable and replaceable; and the metal needle used for printing was a universal dispensing needle, which was also replaceable. During the operation of the micro-mixer, there was a certain hysteresis time between the uniform mixing state of one proportion and the uniform mixing state of another proportion, which was caused mainly by the volume of the mixing chamber, the composition of the material and the length of the transport pipeline. Due to the compressibility of the material and the rigidity of the pipeline and injector used for transportation, the control lag time caused by the fluid pressure in the extrusion process could occur. In order to minimize the impact of the transportation process, in our work, the injection pump and the micro-mixer printer nozzle were installed together on the moving platform to reduce the length of the pipeline. At the same time, the rigid medical nylon semi-rigid tube was adopted to ensure flexible installation and reduce errors caused by the expansion of the hose.

### 2.4. Processing Method of Gel Microspheres

(1) A dynamic mixing nozzle was designed for the preparation of gradient gel microspheres, and corresponding theoretical equations and mathematical models were constructed, respectively, for the mixing and droplet spraying processes in the dynamic mixing nozzle.

(2) Based on the established theoretical mathematical model of mixing and droplet injection, COMSOL Multiphysics multi-physical field simulation software was used to simulate and analyze the mixing and droplet injection processes of the nozzle, respectively. Firstly, the influences of rotor diameter, blade number, rotating speed and fluid viscosity on nozzle mixing efficiency are analyzed. Then the influence of three physical parameters such as pneumatic drive, fluid viscosity and nozzle aperture on droplet injection forming is analyzed. The best parameters are the number of blades is 4, the rotation speed is 120 r/min, the rotor diameter is 22 mm, the concentration of calcium chloride solution is 2% and the concentration of sodium alginate solution is 2%.

(3) Build an experimental platform for preparing gradient gel microspheres and print heterogeneous gel spheres. Firstly, a single and stable gel microsphere was prepared by changing the needle nozzle aperture, pressure drive control and viscosity of sodium alginate solution. Then the color concentration gradient gel microspheres were prepared to prove that the nozzle has good mixing performance and the printing process can realize the change of concentration gradient.

(4) Chondrocytes were used for printing and manufacturing of gel beads with different cell gradients, and chondrocyte gel microspheres containing cell at concentrations of 3 × 10^3^, 3 × 10^4^, 3 × 10^5^, 3 × 10^6^, and 3 × 10^7^ cells/mL were prepared. The prepared gel microspheres were further incubated in an incubator for 48 h at 37 °C saturated with 5% CO_2_. To determine their viability in gel microspheres, the cells were stained with propidium iodide/calcoflavin (live cells were dyed green and dead cells were dyed red). Use YDF-880 fluorescence microscope to observe the cell growth state. The number of cells was measured by blood cell counting plate counting method.

### 2.5. Cell Assay Method

Chondrocytes were used for printing and manufacturing of gel beads with different cell gradients, and chondrocyte gel microspheres containing cell at different concentrations were prepared. The prepared gel microspheres were further incubated in an incubator for 48 h. To determine their viability in gel microspheres, the cells were stained with propidium iodide/calcoflavin (live cells were dyed green and dead cells were dyed red). The cell viability and fluorescence staining data, as well as the viability of cells in gel beads were analyzed.

## 3. The Experimental Process, Results and Discussions

### 3.1. Properties of Sodium Alginate Solution

In practice, the viscosity of sodium alginate solution is an important factor affecting its properties: too high viscosity can cause high fluid pressure and resistance during printing and may lead to blockage of the nozzle or molding failure, whereas too low viscosity can cause cell precipitation when encapsulating cells, which can result in an uneven cell distribution and low gel mechanical strength (because the concentration of materials is too low). Therefore, it is necessary to study factors that affect the viscosity of sodium alginate solution, which can include relative molecular weight, concentration, printing temperature, shear rate, and other factors.

Figure 3I shows the hydrodynamic viscosity at 25 °C of alginate solution at three different concentrations. At a concentration 1%, the viscosity of alginate solution was relatively low with a value of less than 1 Pa·s. With increasing concentration, the viscosity increased continuously, and when the concentration reached 3%, the viscosity curve presented obvious shear thinning; that is, the shear dropped suddenly. The viscosity also decreased with increasing cutting speed. Based on this result, alginate solution at concentrations of 1% or below can be regarded as Newtonian fluid.

When sodium alginate solution is used to encapsulate cells that are used in cell printing, it is necessary to analyze the shear stress in the material in order to understand the damage carried out to cells by the printing process, as the presence of high shear stress can affect the survival rate of cells. The shear stress curves of sodium alginate solution at three concentrations measured by a remoter are shown in Figure 3II. As illustrated, at the shear rate range of 1–1000 s^−1^, the increase of shear rate caused the sheer force of sodium alginate solution to also increase, and the growth rate was faster during this period. Under the same shear rate, the shear stress of sodium alginate solution at a high concentration was higher than that at a low concentration. Compared with that of 2% and 3% sodium alginate solution, the shear stress of 1% sodium alginate solution was much lower under the same shear rate, and the difference was more pronounced at high shear rates. Therefore, in the actual use of the material, when the occurrence of high shear rate is inevitable, reducing the concentration of sodium alginate solution can ensure that the shear force remains low.

The rate of chemical crosslinking between sodium alginate solutions and calcium ions is high. When the droplet is in contact with calcium chloride solution, sodium alginate on the surface can immediately crosslink with the solution to form a thin hydrogel film. At the same time, calcium ions can continue to penetrate into the sodium alginate droplet and continue to crosslink. When the sizes of the microspheres are large, it is more difficult for calcium ions to penetrate into their core. Therefore, in order to study the reaction that take place inside the core of crosslinked sodium alginate, we employed electron microscopic technique to scan and capture the internal ion energy spectrum of gel beads with a diameter of 450 μm, as shown in Figure 4(Ia). During the chemical crosslinking stage, the concentration of calcium chloride solution was 2%, and the volume of the microspheres was sufficient. The time estimated for the crosslinking of 1% sodium alginate solution to be completed was 30 min, and the color of the image was changed to light green, as shown in Figure 4(Ib). After the crosslinking was completed, the pellets were immediately removed and then frozen in liquid nitrogen. The pellets were then placed in a −80 °C freeze-dryer for drying. The electron microscopic images showed that calcium ions in the gel beads had the same energy density after the gel was crosslinked, indicating that alginate hydrogel was completely crosslinked.

In the crosslinking reaction between microspheres and calcium chloride, spherical particles were not always formed. Figure 4II shows electron microscopic images of three different types of extruded calcium alginate hydrogel formed using 2% calcium chloride solution. The comparison showed that under the same conditions, as the concentration of sodium alginate solution increased, the viscosity of the gel increased and its fluidity during crosslinking was worsened. Therefore, after crosslinking, the gel with stretch shape remained in the formation process, causing its shape to deviate from spherical shape. In order to guarantee that the shape of the gel remains spherical, low-concentration sodium alginate solution was selected to prepare the microspheres, and a technology that could slightly pull the liquid drop was adopted.

The viscosity of sodium alginate solution can greatly influence the strength of crosslinked hydrogel, both the tensile strength and gel density. Figure 4III shows SEM images of alginate hydrogel microspheres at different concentrations. The microspheres were dehydrated prior to the scan. As can be seen from the images, the morphology and surface texture of the microspheres varied with the concentration of alginate gel. As the alginate gel concentration decreased (from a to c), the shrinkage of microspheres due to the loss of water became more obvious, and the concave texture on their surface was also more pronounced.

### 3.2. Mathematical Model of Mixing Process and Analysis of Factors Influencing the Mixing Alginate

#### 3.2.1. Mathematical Model Analysis

The mixing process of the dynamic mixing nozzle is similar to the working principle of turbomachinery used in engineering. The mixing process begins as a result of rotation of the stirring rotor. Driven by the stepper motor, the agitating rotor continuously rotates to convert mechanical energy into mixing kinetic energy that can then cause the transfer of fluid material into the rotary domain, to promote the internal fluid to undergo three-dimensional mixing and stirring, and to accelerate the radially and axially mixing of the fluid materials in the mixing chamber. Based on these factors, the mixing of the bio-gel material is fast, efficient and uniform.

The fluid mixed in the dynamic mixing nozzle is assumed to be incompressible turbulent fluid, which can produce centrifugal force and Coriolis force during the mixing process; the internal pressure of the fluid is also isotropic. At a constant temperature, the physical and chemical properties of the fluid are unchanged. The theoretical equations describing the internal mixing flow include turbulence model and viscous fluid control equation.

The turbulence model is shown in Equation (1).
(1)$$R_{e}=\frac{\rho ZD^{2}}{\mu }$$
where: μ is the viscosity of the fluid material; ρ is the density of the fluid; Z is the speed of the rotor; and D is the diameter of the stirring rotor.

According to Equation (1), the larger the Reynolds number, the greater the influence of inertia force on the flow field in the nozzle and the more unstable the fluid flow; as a result, a turbulent and irregular turbulent flow field is formed. The mixing performance of dynamic mixing nozzle can significantly be affected by fluid velocity and fluid kinematic viscosity: the faster the impeller speed, the higher the fluid flow speed, the higher the Reynolds number, and the higher the fluid mixing performance.

In this paper, the k−ε model was used to represent the turbulence model.

The kinetic energy equation ε is shown in Equation (2), and the turbulent dissipation rate equation is illustrated in Equation (3).
(2)$$\begin{array}{l} \frac{\partial \left(\rho k\right)}{\partial t}+\frac{\partial \left(\rho ku_{i}\right)}{\partial x_{i}}=\frac{\partial }{\partial x_{i}}\left[\left(\mu +\frac{\mu_{t}}{\sigma_{k}}\right)\frac{\partial k}{\partial x_{j}}\right]\\ +G_{k}+G_{b}-\rho \varepsilon -Y_{M}+S_{k} \end{array}$$
(3)$$\begin{array}{l} \frac{\partial \left(\rho \varepsilon \right)}{\partial t}+\frac{\partial \left(\rho ku_{i}\right)}{\partial x_{i}}=\frac{\partial }{\partial x_{j}}\left[\left(\mu +\frac{\mu_{t}}{\sigma_{\varepsilon }}\right)\frac{\partial k}{\partial x_{j}}\right]\\ +G_{1\varepsilon }\frac{\varepsilon }{k}\left(G_{k}+G_{3\varepsilon }G_{b}\right)-G_{2\varepsilon }\rho \frac{\varepsilon^{2}}{k}+S_{\varepsilon } \end{array}$$
where: $u_{t}$ is the turbulent viscosity, which can be calculated by Equation (4).
(4)$$\mu_{t}=\rho \frac{k^{2}}{\varepsilon }$$

In Equations (2) and (3): *t* is time; ρ is the fluid density; $u_{i}$ is the mean velocity; $Y_{M}$ is the contribution of fluctuating expansion in compressible turbulence; $G_{1\varepsilon },\;G_{2\varepsilon },\;G_{3\varepsilon }$ are the empirical constants; $\sigma_{\varepsilon },\;\sigma_{k}$ are the Prandtl numbers corresponding to the turbulent kinetic energy and the dissipation rate, respectively; $S_{k}$ and $S_{\varepsilon }$ are the source items, which are defined by users; $G_{k}$ is the production term of turbulent kinetic energy due to average velocity gradient, which can be calculated by Equation (5).
(5)$$G_{k}=\mu_{t}\left(\frac{\partial v_{i}}{\partial x_{j}}+\frac{\partial v_{j}}{\partial x_{i}}\right)\frac{\partial v_{i}}{\partial x_{j}}$$
where: $G_{b}$ is the production term of turbulent kinetic energy due to buoyancy, which can be calculated by Equation (6):(6)$$G_{b}=-\beta g_{i}\frac{\mu_{t}}{\sigma_{\varepsilon }}\frac{\partial T}{\partial x_{i}}$$
where: $g_{i}$ is the component of gravity acceleration in the *i*-th direction; and β is the thermal expansion coefficient.

When the medium is an incompressible fluid, $Y_{M}=0$, $G_{k}=0$, $G_{b}=0$, $S_{k}=0$, and $S_{\varepsilon }=0$.

#### 3.2.2. Simulation of Mixing Process and Analysis

In this paper, COMSOL multi-physical field simulation software is used to simulate the mixing and droplet generation process of the nozzle. The influence on the mixing effect of fluid mixing is analyzed from four design parameters: rotor diameter, number of blades, rotation speed and fluid viscosity. In two-dimensional simulation model, the physical field control grid was selected, and the grid division was automatically generated by the system. User-defined settings for the model were set based on the purposes and requirements. The grid division of the two-dimensional simulation model is shown in Figure 5I.

Firstly, we studied the influence of structural parameters on the mixing performance, while the influence of friction, gravity and surface tension on the fluid and the temperature change in the stirring process were ignored. The inner diameter of the baffle was 30 mm, and the diameter of the rotor was 1.5 mm. The rotor diameter can greatly influence the input of rotational torque, power consumption and mixing time. A reasonable design of the rotor diameter and a reasonable ratio between the rotor diameter and the mixing cavity not only can increase the efficiency of the mixing, but also can help reduce the energy consumption. In this study, the number of blades was 4, and the rotation speed Z was 120 r/min. As shown in Figure 5II, at t = 5 s, the rotor diameter was set at 10, 12, 16, 18, 22, and 24 mm.

Analysis of the simulation results showed that at t = 5 s and with rotor diameters of 22–24 mm, the mixing reached the uniform mixing state, and with the rotor diameters of 14–16 mm, trace substances were approaching the state of uniform mixing. In addition, when the rotor diameters were between 10 and 12 mm, trace substances only began to release and were largely concentrated in the low-speed flow region, an obvious indication of low mixing efficiency. The above observations may be due to the following reasons. At the same speed, the rotor diameter determines the volume force of the flowing fluid. At a certain range of speed, the larger the rotor diameter, the greater the volume force, the higher the velocity, the higher the turbulent kinetic energy, the shorter the mixing time, and the better the mixing. When the rotor diameter exceeded 20 mm, the distance between the top edge of the rotor blade and the inner wall of the mixing chamber decreased, and the flow state was greatly restricted, all of which are not conducive to the diffusion of trace substances, causing the mixing efficiency to significantly reduce. Based on the above, the appropriate impeller diameters should be between 22 and 24 mm. To ensure high mixing efficiency and low power consumption, the stirring rotor with a diameter of 22 mm was selected.

The proper number of blades can promote the rotor blades to continuously and effectively produce strong shear effect on the solution, further break the liquid particles, accelerate the separation diffusion and flow of the fluid, and improve the mixing efficiency. At the same velocity, the continuous, strong shear action of the fluid material cannot occur when the blades are small. When the number of the designed blades is large, the spacing between blades is very low, thus can lead to the accumulation of fluid materials between the blades, which is not conducive to flow mixing. Based on the above, to study the influence of the number of blades on the mixing effect, the number of blades was optimized. The diameter of the rotor was set at 22 mm, the rotation speed was set to 120 r/min, and the fluid material was sodium alginate solution. The fluid mixing process was simulated at the number of blades of 4, 6, 8, and 10, and at time = 4.1 s, and the results are shown in Figure 5III.

It can be seen from the Figure 5III that at time = 4.1 s, the concentration distribution of trace substances in the four states was similar, but the magnitude and scope of turbulent kinetic energy of rotors at different blade numbers were greatly different. When the number of rotor blades was 8, the maximum turbulent kinetic energy was 1.68 × 10^−7^ m^2^/s^2^, where as the minimum energy was 8.36 × 10^−9^ m^2^/s^2^. Eight rotor blades generated suitable stirring effect and strong turbulent kinetic energy, thus had high fluid mixing efficiency.

The rotational speed determines the energy input from the stepping motor to the rotating fluid, which can directly affect the mixing of fluid in the mixing chamber and the mixing time of the dynamic mixing nozzle. Reasonably setting the rotational speed of the rotating blades can enhance the mixing of gel materials, reduce the mixing time, and improve the mixing efficiency, as well as can reduce the power consumption of stepper motors and save resources. Based on the above analysis, the optimal setting should be as follows: rotor diameter, 22 mm; number of blades, 8; width of cavity baffle, 1.5 mm; and number of baffles, 4. Figure 5IV show mixing at rotation speeds of 20, 50, 70, 100, 120 and 150 r/min, and at time = 4.1 s.

Analysis of the simulation results showed that when the impeller speeds were 20–50 r/min, the fluid mixing was too slow, the diffusion of trace substances was low, and the mixing was not satisfactory, as it remained at the initial release position for a long period of time. With increasing rotor speed, the speed of fluid gradually increased and the transfer of trace substances gradually became accelerated. However, when the speeds were between 70 and 100 r/min, the trace substances were concentrated at the root of the impeller over a large area. This is likely because of the speed is too low, and the speed at the high-speed shear zone at the top of the blade is not sufficiently high to transfer trace substances to other zones. As a result, trace substances may prematurely enter the gap between rotor blades (the low-speed shear zone), and the mixing speed is so slow that trace substances are concentrated in a small area and remain at high concentrations. When the rotating speeds were between 100 and 150 r/min, the mixing was drastically improved. However, the concentration distribution and the mixing efficiency were not further improved at these speeds, and the change trend was nearly flat. Therefore, considering the power consumption of the motor and other factors, a higher rotating speed is better. The optimal speed of the active mixing nozzle was about 120 r/min.

### 3.3. Effects of Different Factors on Formation of Gel Microspheres

The crosslinking of alginate gel microspheres is complex and variable. The morphology of microspheres can depend on the volume of sodium alginate solution and the alginate gel droplet size. As described above, the droplet size could be controlled by the size of the needle and the driving pressure. Thus, the effects of three factors including pinhead size, driving pressure and concentration of alginate solution on the morphology of gel microspheres were determined.

Figure 6I shows the morphology of gel microspheres generated from 2% sodium alginate and needles with different diameters. The diameters and sizes of the gel microspheres were different: the larger the diameter of the needle, the larger the volume of the gel microsphere.

The morphology of the gel microspheres formed from 2% alginate and the needle with a size of 0.9 mm is shown in Figure 6II. Under different air pressures, the gel microspheres had different efficiencies and morphologies. At a lower air pressure, the gel microspheres dropped slowly, but the overall morphology was relatively complete, and the tail gel filaments were relatively shorter. At a higher air pressure, the gel microspheres had higher efficiency, but also had elongated filamentous structures at their rear.

Gel microspheres formed from 1%, 2%, and 3% alginate, 2% calcium chloride solution, and the needle with a diameter of 0.9 mm are depicted in Figure 6III. As can be seen, the concentration of sodium alginate solution could greatly influence the morphology of gel microspheres: the higher the viscosity of alginate solution, the lower the fluidity. During the formation of gel droplets, a slim filamentous gel structure could easily be formed at the tail of the droplet. Alginate at a low concentration generated spherical gel microspheres. Multiple gel microspheres could easily bind to one another to form microspheres with a larger size, and these large-size gel microspheres were not sufficiently round and could easily become elongated.

Alginate solution was dyed with two different colors (red and blue inks) at the same concentration, and the contrast between the two was distinct. The amount of red color was controlled in order to obtain gel beads with different color gradients during printing; as a result, the colors of the gel microspheres gradually changed from transparent to blue, purple, pink, and red. The intensity of the colors is closely linked to the stirring speed, mixing time and the amount of the two colors in sodium alginate solution that was added into the mixing chamber. The stepper motor was turned on to stir at a speed of 60 r/min. Using a 60-mL syringe, pure blue sodium alginate solution was introduced into the left inlet at a speed of 0.2 mm/s. Pure blue sodium alginate droplets were then produced from the needle, but after pure red sodium alginate solution was further introduced at a speed of 0.2 mm/s, the produced droplets gradually became purple.

Figure 7I printed the gradient change of the material with the geometric path of the Archimedes spiral. By calculating the color change of the material along the spiral curve, the mixing ratio of the two materials at a certain time can be determined. In this experiment, the gel spheres we need are not homogeneous, and the cell content of each part of a gel sphere is different. In this way, a certain mixing ratio gel ball needed for subsequent experiments is selected by controlling the time.

The microsphere was printed according to the principle of Figure 7I,II shows gel microspheres dyed with red and blue at different ratios (0:4, 1:3, 2:2, 3:1, and 4:0). As can be seen, the color density of the gel microspheres gradually changed from pink to dark red, purple, dark blue and pale blue.

### 3.4. Printed Gel Microspheres Containing Cells at Different Densities

Chondrocyte solution containing cells at different densities were prepared. The cell suspension was withdrawn using a syringe and then fixed on the micro-injection pump. The injector and the nozzle inlet were connected to both sides of the nozzle through a tube to ensure the survival of the cells. The motor speed was set to a reasonable speed about 120 r/min. To obtain alginate gel droplets containing cells at different concentrations, the feed speed of the micro-injection pump was adjusted. Finally, the gel microspheres with different cell gradients were formed in a cell incubator at 37 °C saturated with 5% CO_2_.

Chondrocytes were used for printing and manufacturing of gel beads with different cell gradients, and chondrocyte gel microspheres containing cell at concentrations of 3 × 10^3^, 3 × 10^4^, 3 × 10^5^, 3 × 10^6^, and 3 × 10^7^ cells/mL were prepared. The prepared gel microspheres were further incubated in an incubator for 48 h. To determine their viability in gel microspheres, the cells were stained with propidium iodide/calcoflavin (live cells were dyed green and dead cells were dyed red). Use YDF-880 fluorescence microscope to observe the cell growth state. The number of cells was measured by blood cell counting plate counting method.

The cell viability and fluorescence staining data, as well as the viability of cells in gel beads were analyzed. According to the data shown in Figure 8I, when the concentration of cells in the gel microspheres was less than or equaled to 3 × 10^6^ cells/mL, the cell survival rate was relatively high, up to 90%. By contrast, as the cell concentration was increased, the cell survival rate became much lower, the survival rate is about 65%. This could be due to that when the gel microspheres had the same sizes and when the cell concentration was low, the cells received more nutrients, causing their growth rate to be relatively higher; as a result, their survival rate was higher. On the other hand, when the cell concentration was high, the cells competed for nutrients, causing insufficient supply of growth materials, resulting in a large amount of cell death.

Cell Gel Microspheres cultured for 1d–5d were placed under YDF-880 fluorescence microscope to observe the cell growth state. As shown in Figure 8II, the viability of cell at a concentration of 3 × 10^6^ cells/mL in the gel microspheres was observed over 5 days. The growth of cells in the gel microspheres was relatively high. The survival rate of cells on the fifth day was significantly higher than that of cells on the first day. This indicates that alginate gel microspheres are suitable for three-dimensional culture of chondrocytes.

## 4. Conclusions

This paper presents the fabrication of heterogeneous gel microspheres and biological printing to provide some insights into the printing processes and the construction of complex cell microenvironment in vitro. The following were conducted and observed:

1. The main factors affecting the materials including hydrodynamic viscosity, shear stress and crosslinking characteristics of sodium alginate solution (three different concentrations) were assessed. The experiments showed that 2% sodium alginate was most suitable for constructing gel microspheres.

2. A new type of micro-mixing nozzle based on dynamic stirring was used to print the heterogeneous gel microspheres. The principle of active mechanical stirring was employed to achieve the rapid mixing of multi-component gel materials; and it was also used as a micro-mixing nozzle for cell printing.

3. The survival rate of cells in gel microspheres was also investigated, and the results showed that using our technique, cells had very low death rate (≤20%) according to the data shown in Figure 8 and could exchange substances within the microspheres.

## Figures and Tables

**Figure 1 materials-15-02305-f001:**
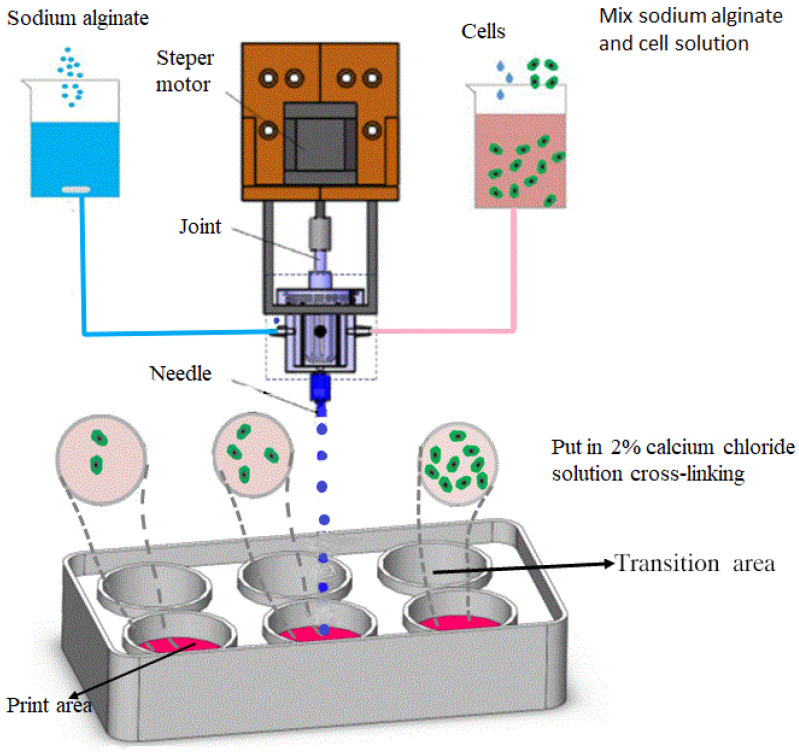
Printing scheme of cell density gradient gel microsphere.

**Figure 2 materials-15-02305-f002:**
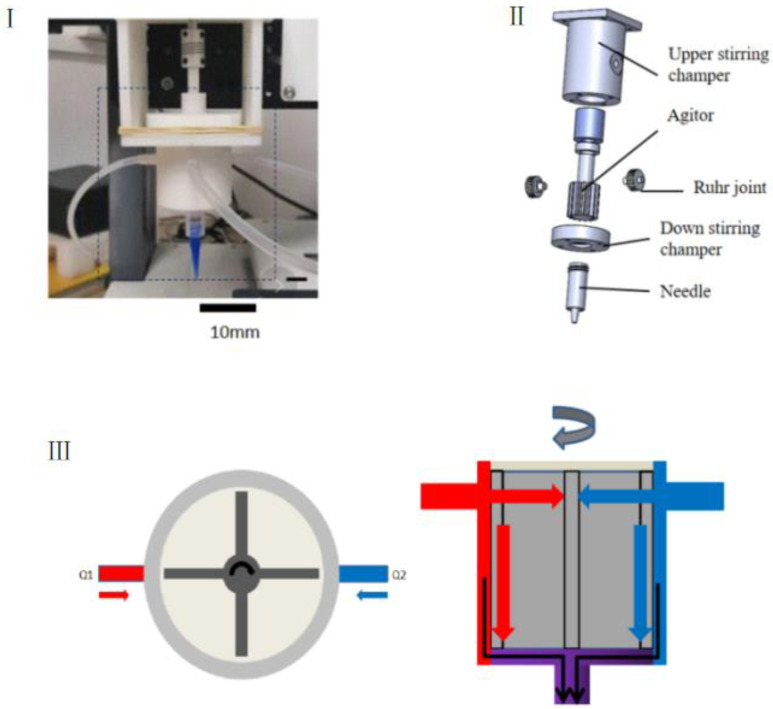
Gel Microspheres setup with dynamic micro-mixer nozzle: (**I**) physical model; (**II**) detailed structure; (**III**) dynamic mixing nozzle mixing process principle diagram.

**Figure 3 materials-15-02305-f003:**
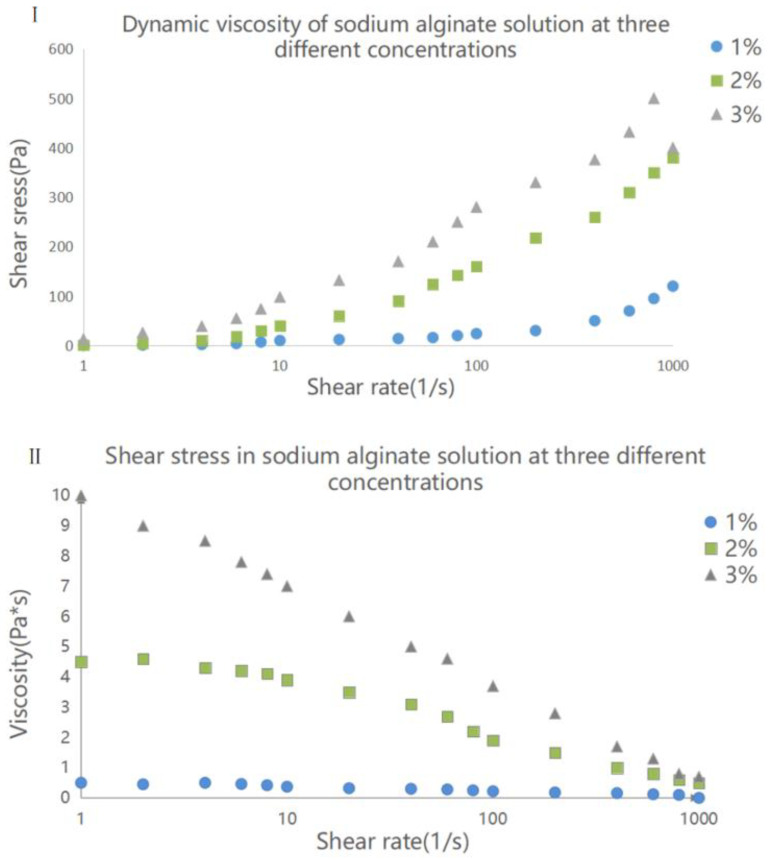
(**I**) Dynamic viscosity of sodium alginate solution at three different concentrations. (**II**) Shear stress in sodium alginate solution at three different concentrations.

**Figure 4 materials-15-02305-f004:**
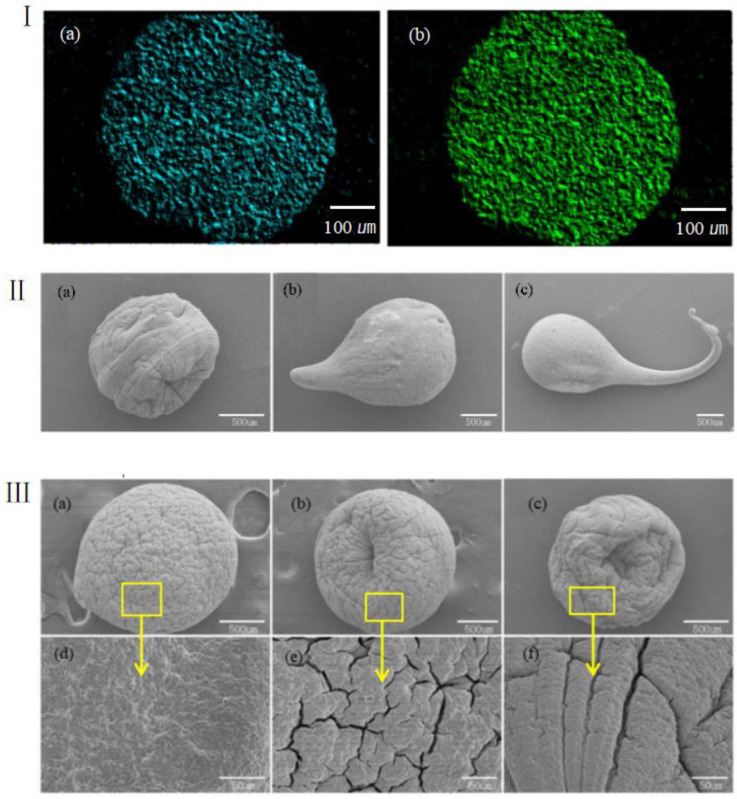
(**I**) Ion distributions in sodium alginate microspheres (**a**) before and (**b**) after crosslinking. (**II**) SEM images of different shapes of calcium alginate gel particles at three concentrations: (**a**) 1% sodium alginate solution; (**b**) 2% sodium alginate solution; and (**c**) 3% sodium alginate solution. (**III**) SEM images of the surface of calcium alginate gel particles at three different concentrations: (**a**) 2% sodium alginate after crosslinking and freeze-drying; (**b**) 1% sodium alginate after crosslinking and freeze-drying; and (**c**) 0.5% sodium alginate after crosslinking and freeze-drying. (**d**–**f**) Close-up images of (**a**–**c**) at 800 times magnification.

**Figure 5 materials-15-02305-f005:**
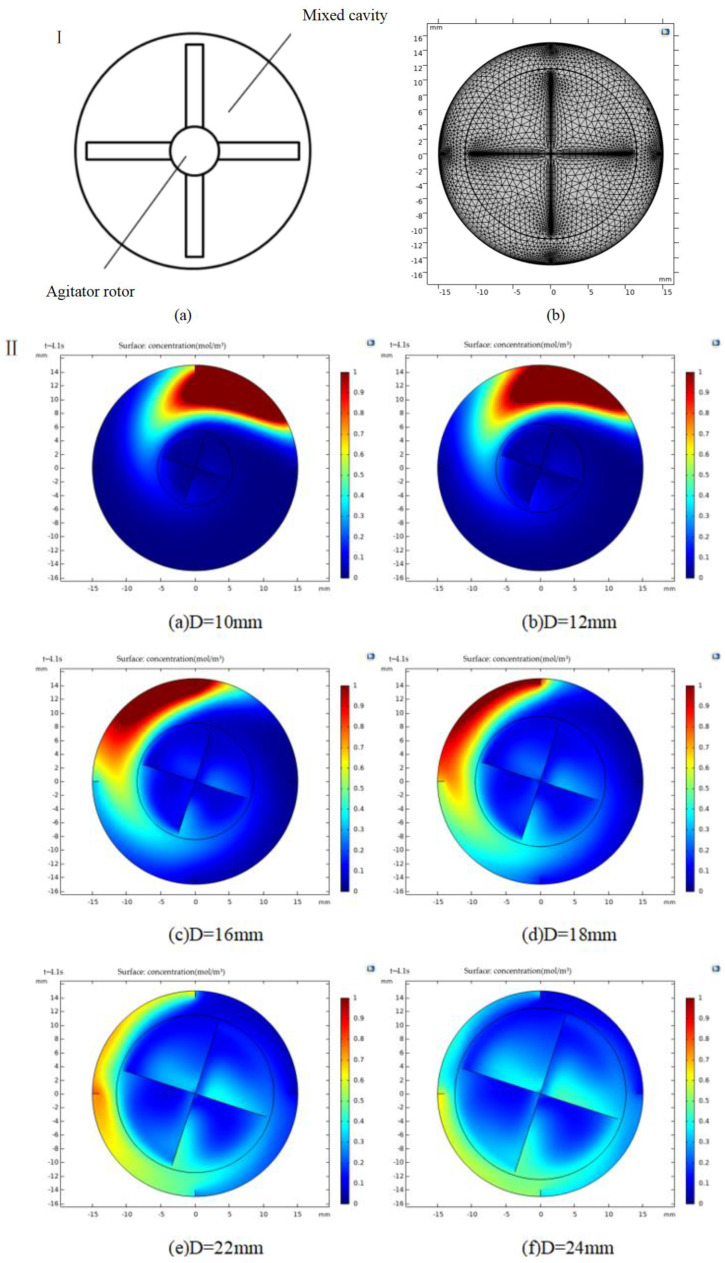
(**I**) Simulation model of dynamic stirring rotor: (**a**) two-dimensional structure and (**b**) grid model. (**II**) Fluid mixing diagram at different rotor diameters (**a**–**f**) and t = 4.1 s. (**III**) Fluid mixing diagram under different blade numbers (**a**–**d**) at t = 4.1 s. (**IV**) Fluid mixing diagram at different rotation speeds of (**a**–**f**) and at t = 4.1 s.

**Figure 6 materials-15-02305-f006:**
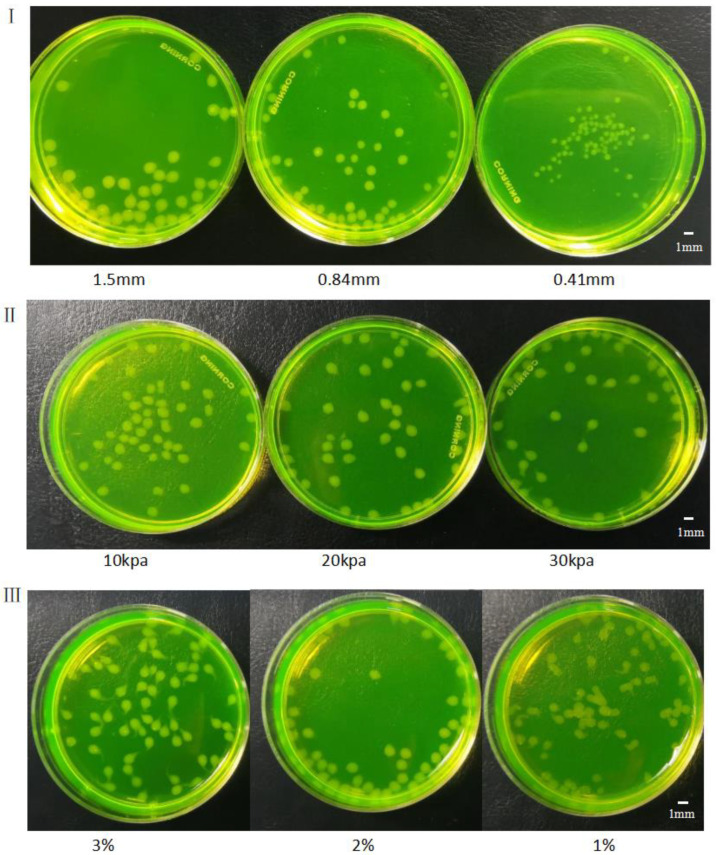
Gel microspheres formed (**I**) through needles with diameters of 1.5 mm, 0.84 mm and 0.41 mm. (**II**) under different driving pressures: 10 kPa, 20 kPa and 30 kPa. (**III**) from alginate at concentrations of 3%, 2% and 1%.

**Figure 7 materials-15-02305-f007:**
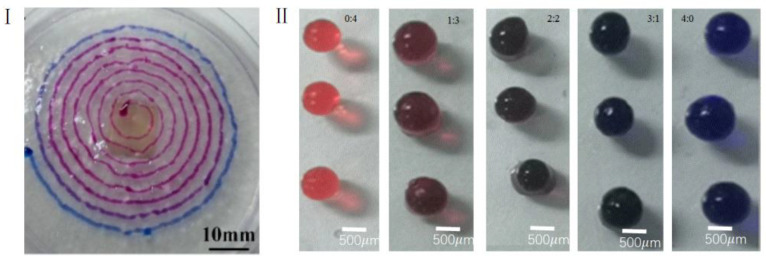
(**I**) gradient change of the material with the geometric path of the Archimedes spiral; (**II**) gel microspheres dyed with red and blue at different ratios (0:4, 1:3, 2:2, 3:1, and 4:0).

**Figure 8 materials-15-02305-f008:**
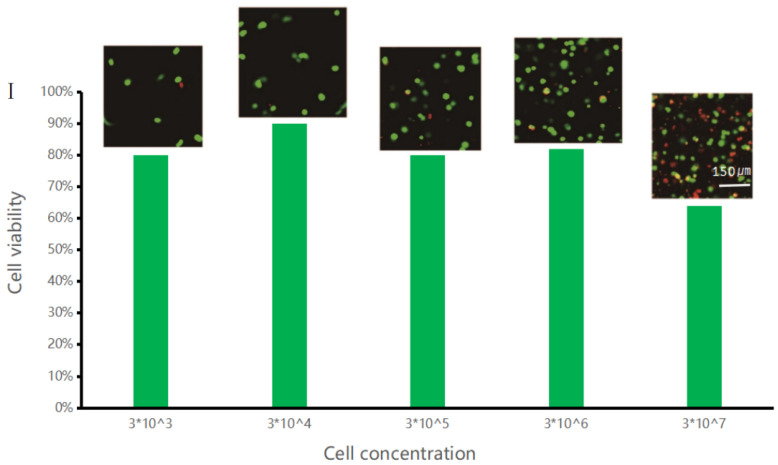
(**I**) LIVE/DEAD cell assay of cells at different concentrations. (**II**) Survival rate of cells in gel microspheres at a concentration of 3 × 10^6^ cells/mL observed over 5 days.

**Table 1 materials-15-02305-t001:** List of experimental reagents.

The Name of the Reagent	Specifications	Company
Fetal bovine serum (FBS)	Biological level	Sigma-Aldrich (Germany)
DMEM medium	Cell culture level	Sigma-Aldrich (Germany)
Penicillin-Streptomycin solution	Biological level	Sigma-Aldrich (Germany)
0.25% Sterile PBS buffer	Cell culture level	Sigma-Aldrich (Germany)
Phosphate-buffered saline (PBS, pH: 7.4)	Cell culture level	Sigma-Aldrich (Germany)
75% Medical alcohol	Analytical Reagent	Merck (Germany)
Anhydrous calcium chloride	Analytical Reagent	Fluka (Switzerland)
Sodium alginate	Analytical Reagent	Fluka (Switzerland)
DMSO		Sigma-Aldrich (Germany)
Calcein	K305-1000	Sigma-Aldrich (Germany)
Propidium iodide	DMEM/F-12, HEPES	Merck (Germany)
Deionized water		

## Data Availability

The study did not report any data.

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
