# Peer review of "Gradient Printing Alginate Herero Gel Microspheres for Three-Dimensional Cell Culture"

_materials, 2022, doi:10.3390/ma15062305_

Round 1

Reviewer 1 Report

The manuscript itself is very interesting and pretty sure a wide audience will be interested in it after a major revision.

There are some crucial problems with the manuscript:

1, The introduction itself is not a coherent chapter rather just a listing of the results from different papers, but it is hard to figure out what is the connection between the manuscript and the referred papers. Thus it would be nice to rewrite the introduction in a much more fluent way instead of listing one by one what can we find in the literature.

2, In the materials chapter, it is more an introduction than a material, no details are mentioned in that chapter (2.1). So please delete that part and write one by one the used materials instead of writing a literature summary.

3, Chapter 2.3 does not contain any details about the experimental setup, thus nobody can repeat that experiment based on the written sentences. Add a properly detailed paragraph about the experimental conditions ( how many cells were used in 1 container, what amount of microliter was used from the buffer and so on).

4, Refer the figures in the correct order (Fig 2C was referred first). And use the same reference type, once the authors used Figure than Fig.

5, In chapter 2.3 there are not any exact detail about the experiment again...no ammounts no mixtures.

6, Cell assay again no information about the number of cells, how the data analization happened and so on, no details.

7, Please make Figure 3 in a proper way with the same font size for the axes and the same labels for the same data series for a better understanding. The labels on the second graph are not very proper.

8, The 3.2 chapter is a result or a method? It is not clear that the equations are figured out by the authors or such as the first one it is well known from the literature, thus just a method. Because in the second case the authors have to reorganize that section.

9, In the method part, there is no word about any simulations...please add the exact program what was used for the simulation, the conditions for the calculations and all the details which is important for the readers.

10, The quality of the Figure 6 is not good enough to summarize any morphology change or effect on the beads. Thus show some better images or delete that part of the paper.

11, The relevance of Figure 7 is not clear, it is nice, but do not really understand why it is important. Please add some longer explanation of that to get a better understanding to the readers.

12, In chapter 3.5 a lot of experimental conditions are mentioned which is not a result, but a method part of a manuscript. Please remove it from the result and add it to the method section.

13, In Figure 8 what is the 100%? What was the reference? Which assay was used for the viability test?

Author Response

Dear Editors and Referees,

Thanks for your letter and for the referees’ comments concerning our manuscript entitled “Gradient Printing Alginate Herero Gel Microspheres for Three-Dimensional Cell Culture” (Manuscript ID: materials-1589696). Those comments are all valuable and very helpful for revising and improving our paper, as well as the important guiding significance to our researches. We have studied these comments carefully and have made correction which we hope meet with approval.

Reviewer 2 Report

The paper presented by the authors describe the fabrication of heterogeneous gel microspheres, and considers the main factors affecting the materials used, demonstrated the use of a new type of micro-mixing nozzle based on dynamic stirring and investigated the survival rate of cells in the gel microspheres.

Although the field is very relevant, and the method could be very useful to researchers in the field, the presentation of the manuscript is not clear and to the point, and details of the methods are often ambiguous and difficult to follow which is a problem for an intrinsic “method” paper.  With some rewriting, it can be much improved. Language editing will also be necessary.

General comments:

  • The background is a bit too much of a literature review article, and does not highlight the specific reasoning behind the proposed method enough.
  • The method section is written more like a background or a text book and does not clearly describe the experimental procedure in an unambiguous way. It should be rewritten.
  • Section 3.2 and 3.3 are not unrelated and should be merged.
  • Many times “cell” is written as singular where it should be plural “cells”
  • You refer to both L929 mouse fibroblasts and chondrocytes? Were both used? Please also clarify the source of the cells and the culture conditions and medium composition used to culture the cells.
  • Was the effect of shear forces on cell viability evaluated?

Specific comments:

L120 – incomplete sentence

L175 – Do you not mean gel-containing cells?

L284 – What do you mean by “the volume of the microspheres was sufficient”?

L495 – A reasonable speed? Clarify?

Figure 8 – If it is a live/dead assay, why refer in the figure to survive/die?

L537 – Please expand on “low death rate”.  Can you quantify this in terms of a percentage of total cells?

Author Response

(The authors gave the same response as above.)

Round 2

Reviewer 1 Report

After reading the paper, the reviewer still has the same opinion:

"1, The introduction itself is not a coherent chapter rather just a listing of the results from different papers, but it is hard to figure out what is the connection between the manuscript and the referred papers. Thus it would be nice to rewrite the introduction in a much more fluent way instead of listing one by one what can we find in the literature."

The reviewer did not see any changes in the introduction, so please rewrite it before upload a newer version.

Please cite the following paper also: https://www.mdpi.com/2310-2861/8/2/65

2, In the materials chapter, please refer to which company produces the chemicals (by the way Reagent not Reagen)

3, Chapter 2.3 does not contain any details about the experimental setup, thus nobody can repeat that experiment based on the written sentences. Add a properly detailed paragraph about the experimental conditions ( how many cells were used in 1 container, what amount of microliter was used from the buffer and so on). Please do not refer methodology in the Results and Discussion part, thus put all the experimental details to the Method section from the 3.4 chapter.

4,In Figure 8 I what is the 100%? What was the reference? Please do not link the points, because it shows false results. These are independent experiments, maybe a bar graph would fit better like the same way as it is in the II part.

Author Response

This manuscript is a resubmission of an earlier submission. The following is a list of the peer review reports and author responses from that submission.